# An In Vitro Evaluation of Selenium Nanoparticles on Osteoblastic Differentiation and Antimicrobial Properties against *Porphyromonas gingivalis*

**DOI:** 10.3390/nano12111850

**Published:** 2022-05-28

**Authors:** Jason Hou, Yukihiko Tamura, Hsin-Ying Lu, Yuta Takahashi, Shohei Kasugai, Hidemi Nakata, Shinji Kuroda

**Affiliations:** 1Department of Oral Implantology and Regenerative Dental Medicine, Tokyo Medical and Dental University, Tokyo 113-8510, Japan; j_hou.irm@tmd.ac.jp (J.H.); luirm@tmd.ac.jp (H.-Y.L.); shoheikasugai@gmail.com (S.K.); 2Department of Dental Pharmacology, Tokyo Medical and Dental University, Tokyo 113-8510, Japan; tamu.hpha@tmd.ac.jp; 3Dental Hospital Clinical Laboratory Division, Tokyo Medical and Dental University, Tokyo 113-8510, Japan; y57ydlab@tmd.ac.jp

**Keywords:** selenium, nanomaterial, osteoblastic differentiation, antimicrobial, peri-implantitis, *Porphyromonas gingivalis*

## Abstract

Despite numerous treatment methods, there is no gold standard for the treatment of peri-implantitis—an infectious peri-implant disease. Here, we examined selenium nanoparticles (SeNPs) at a wide range of concentrations to investigate their cytotoxicity, regulation of osteoblastic differentiation, and assessed the antibacterial effect against *Porphyromonas gingivalis*. SeNPs (mean size: 70 nm; shape: near-spherical; concentration: 0–2048 ppm) were tested against the MC3T3-E1 osteoblast precursor cell line and *P. gingivalis* red complex pathogen. Reverse transcriptase quantitative polymerase chain reaction (RT-qPCR) analysis was used to evaluate the bone morphogenetic protein 2 (BMP-2) signaling pathway. SeNPs at concentrations of 2–16 ppm showed no obvious cytotoxicity and promoted good mineralization and calcification. SeNPs at concentrations 64 ppm and below influenced gene expression promoting osteoblastic differentiation, whereas at high concentrations inhibited the expression of Runt-related transcription factor 2 (Runx2). The growth of *P. gingivalis* was significantly inhibited at SeNP concentrations of more than 4 ppm. SeNPs at low concentrations promoted osteoblastic differentiation while strongly inhibiting peri-implantitis pathogen growth. This study represents one of the few in vitro assessments of SeNPs against a red complex pathogen and the regulatory effect on osteoblastic differentiation. The findings demonstrate SeNPs could potentially be used for future application on implant coating.

## 1. Introduction

In recent years, research and developments on nanoparticles have been widely studied due to the investigation of their therapeutic properties and progressive application in modern medicine [1,2,3,4]. Nanoparticles have been incorporated on the surface of implants to improve the durability of implants by providing protective properties. An ideal implant procedure requires the implant to securely anchor with the bone, thus inducing osseointegration, as discovered by Brånemark et al. in 1969 [5]. However, biological complications can affect the process and integrity of osseointegration. Peri-implantitis is a challenging health complication, which is common in oral implant restorations. This disease substantially reduces the success and longevity of restorations through the destruction of the alveolar bone, thus destroying the osseointegrated connections [6]. The etiology of peri-implantitis is multifactorial, with the primary factor being diverse microbiota compositions associated with periodontal pathogens [7]. Studies have indicated that periodontitis-associated bacteria can colonize peri-implant pockets within a week of implantation [8]. Individuals who harbor periodontal pathogens are more susceptible to peri-implantitis [8]. Several treatments have been designed for the debridement of bacterial and biofilm formation, including non-surgical mechanical methods such as ultrasonic and laser devices, chemical methods using antiseptic and antibiotic agents, and surgical methods exposing the subgingival region [9,10]. However, the treatment and decontamination of the implant surface do not establish an ideal prognosis for re-apposition to the bone; thus, it is difficult to completely achieve re-osseointegration [9,11]. Therefore, the use of agents on the surface of the implant to prevent bacterial proliferation and biofilm formation is an attractive method of decreasing the risk of such biological complications.

Recent advances in nanotechnology have led to the development of various novel nanomaterials with several industrial and biomedical applications. Silver nanoparticles have gained considerable attention in the field of medicine; they can be used in the form of an implant coating owing to their chemical stability and antibacterial and biocompatible properties [12,13,14,15]. However, the cytotoxicity of silver nanoparticles in both in vitro and in vivo models has raised concerns in clinical applications [12,15,16]. Selenium is a trace element that is essential for the appropriate functioning of the human body. The anti-oxidative and pro-oxidative effects of selenium help to reduce oxidative stress (ROS) and create an equilibrium between oxidation–reduction chemical reactions in a large array of tissues and cell types, including those involved in natural and acquired immune responses [17]. In recent years, there has been growing interest in the preparation and study of selenium nanoparticles (SeNPs) [18,19,20]. SeNPs have shown antibacterial activity against Staphylococcus aureus, methicillin-resistant S. aureus, and Staphylococcus epidermidis [20,21,22], and have no or low cytotoxicity against fibroblasts [19,20]. However, previous studies on SeNPs have not performed any in-depth analysis to establish SeNPs’ suitability in clinical studies. Even in its early stage, relevant data on SeNPs against osteoblastic differentiation and peri-implantitis pathogens are extremely limited. In the field of orthopedics and dentistry, an ideal implant surface modification should exhibit both antibacterial and osteoblastic properties. Thus, with none or very few existing studies, we aimed to provide new data on the effects of SeNPs concentrations with experimental assessments investigated based on these ideal properties.

The objective of this study is to establish the potential of SeNPs for future application as a surface modifier for dental implants by firstly investigating the osteoblastic differentiation effects and antibacterial properties of SeNPs at a wide range of concentrations through the examination of cytotoxicity, and antibacterial effect in an in vitro assessment. We hypothesize that SeNPs at a higher concentration can exert optimal antibacterial effect while promoting osteoblastic differentiation with minimal cytotoxicity, thus, potentially highlighting possible implementation in the future as an implant coating in both medical and dental fields. The MC3T3 osteoblast precursor cell line was used to observe the osteoblastic differentiation effects, as it serves as a reliable alternative to the primary human osteoblast in an in vitro cell model. *Porphyromonas gingivalis*, a red complex bacterium in periodontitis, was selected for this study as it is strongly associated with peri-implantitis.

## 2. Materials and Methods

### 2.1. Selenium Nanoparticle Preparation

SeNPs were purchased from NanoShel LLC (Wilmington, DE, USA). According to the manufacturer, SeNPs were synthesized by reducing sodium selenite (Na_2_SeO_3_(H_2_O)_5_) with ascorbic acid and stabilized with polysorbate 20. The exact quantities of reagents were not specified by the manufacturer. Nevertheless, using commercial SeNPs will provide consistent purity and enable to use precise concentrations. For preparations of varying concentrations, SeNPs in powder form were suspended in different media depending on the respective assays (cytotoxicity, osteoblastic differentiation, and antibacterial assay), and then sonicated (Branson Yamato 2510; Marshall Scientific LLC Hampton, NH, USA) until the nanoparticles were fully submerged and segregated in the medium. Subsequently, the suspended SeNPs were thoroughly mixed and serially diluted. SeNPs at a wide range of concentrations were prepared—0 (Control), 2, 4, 8, 16, 32, 64, 128, 256, 512, 1024, and 2048 parts per million (ppm) (ppm = mg/L). The SeNP mixtures of different concentrations were stored until further testing.

### 2.2. Characteristics of Selenium Nanoparticles

The nanoparticles were imaged using a transmission electron microscope (TEM; H-7100/XR81, Hitachi, Tokyo, Japan) with an accelerating voltage of 75 kV, emission current of 1520 μA, and resolution of 0.38 nm. A drop of nanoparticle suspension was added onto a TEM grid and imaged with TEM H-7100. The size, morphology, and aggregation level of the SeNPs were observed. The size of the SeNPs was additionally examined using a Melvern Zetasizer Nano ZS (Malvern Instruments Inc., Malvern, UK). The zeta potential test was performed in water to observe the charge and stability of the nanoparticles. To examine the influence of the cell growth medium on the nanoparticle stabilities, SeNPs were suspended with minimum essential medium (MEM) with 10% fetal bovine serum (FBS) and 1% penicillin-streptomycin (MEM + 10% FBS) and observed using a Malvern Zetasizer Nano ZS (Malvern Instruments Inc., Malvern, UK) at 25 °C.

### 2.3. Cell Culture Preparation

MC3T3-E1 osteoblast precursor cell line (RIKEN Bioresource Center, Tsukuba, Japan) was used for all subsequent cytotoxicity and osteogenesis experiments. MC3T3 cells were prepared in a flask with MEM + 10% FBS, and then cultured at 37 °C with 5.0% CO_2_ for 24–48 h to achieve adequate cell density. The seeded MC3T3 cells in the flask were washed with phosphate-buffered saline (PBS) and prepared for removal using PBS (9 mL) + EDTA (1 mL). After removing the cells from the flask, FBS (1 mL) was added, and the cells were centrifuged at 1500 rpm for 5 min and prepared at the desired density following the standard protocol for the EVE automated cell counter (NanoEntek, Waltham, MA, USA). MC3T3 cells at a density of 100,000 cells/mL were added into plates with appropriate numbers of wells, according to the experiment, and re-cultured for 24 h. SeNPs at 12 different concentrations were suspended in MEM + 10% FBS and allocated to each well of the seeded MC3T3 cells. The samples were cultured at 37 °C with 5.0% CO_2_ for the indicated period for each assay.

#### 2.3.1. Proliferation and Cell Viability CCK-8 Assay

The cytotoxicity of the SeNP solutions of different concentrations was observed by colorimetric assays to determine the number of viable cells in the proliferation and cytotoxicity assays on days 3, 5, and 7. According to the Cell Counting Kit-8 (CCK-8) (Dojindo, Kumamoto, Japan) system, CCK-8 solution was added to each sample in a 96-well plate and cultured at 37 °C for 60 min. After the indicated time, cell proliferation was quantified using an iMark Microplate reader (BIO-RAD, Hercules, CA, USA) at an absorbance of 450/620 nm [23].

#### 2.3.2. Alkaline Phosphatase and Bicinchoninic Acid Protein Assay

The alkaline phosphatase (ALP) concentration was measured to validate the presence of osteoblast cells and the formation of new bone. MC3T3 cells were seeded into 24-well plates that provided adequate cell volume for both ALP and bicinchoninic acid (BCA) protein assays and were appropriate for cell scraping. After 24 h of culture, SeNPs in osteogenic differentiation medium consisting of MEM + 10% FBS, β-glycerol phosphate (10 nM), ascorbic acid (50 μg/mL), and dexamethasone were added to the seeded MC3T3 cells, which were then cultured for 3, 5, and 7 days. After the corresponding culture period for each 24-well plate (3, 5, or 7 days), the cells were washed with 0.9% sodium chloride and scraped in the same direction. According to the manufacturer’s protocol, 20 μL from each sample was displaced into a 96-well plate with 100 μL of working reagent (6.7 mmol/L p-nitrophenyl phosphate disodium + 2.0 mmol/L MgCl_2_, 0.1 mol/L carbonate buffer) allocated into each well. The wells were cultured for 15 min before measuring the ALP amount using an iMark Microplate reader (BIO-RAD, Hercules, CA, USA) at 405 nm [23].

Using the resulting ALP amount values, the p-nitrophenol (nmol) value was extrapolated with a standard curve created following the manufacturer’s protocol. The relative ALP activity was calculated using the following equation: ALP activity = ((p-nitrophenol/15) × 10)/BCA protein. A BCA protein assay kit was used to quantify the total protein for the calculation of relative ALP activity. According to the manufacturer’s protocol, cells (25 μL) pretreated with SeNPs of different concentrations, obtained from the previous 24-well plate, were transferred to a 96-well plate. The working reagent (200 μL) from the BCA protein kit was added to each well and mixed thoroughly for 30 s. The samples were incubated at 37 °C for 30 min. After incubation, the absorbance was measured with the iMark microplate reader at 570 nm.

#### 2.3.3. Alizarin Red Staining

Alizarin red staining was performed to analyze the deposition of calcium containing osteogenic differentiated cells. Similarly, MC3T3 cells at 100,000 cells/mL were seeded in a 24-well plate along with SeNPs at different concentrations. The osteogenic differentiation medium was changed every 48 h. After 21 days of culture, the cells were thoroughly washed with PBS and fixed with 10% formalin for 15 min. Subsequently, the cells were washed with double distilled water (dd-H_2_O) just before staining with 1% Alizarin Red S solution for 5 min and then washed twice with dd-H_2_O. The excess water was removed, after which the samples were left to air-dry before analysis [24].

### 2.4. Preparation for Reverse Transcriptase Quantitative Polymerase Chain Reaction (RT-qPCR)

To observe the expression of genes in the osteoblastic pathway, cells prepared with SeNPs at different concentrations were cultured in a 6-well plate for 3 days and another 6-well plate for 7 days. TRIzol reagent was added to each well. Each well was thoroughly scraped in the same direction, and TRIzol containing the cells (1 mL) was collected and transferred into sterilized Eppendorf tubes. Subsequently, chloroform (0.3 mL) was added, and samples were vigorously mixed and left at room temperature (20–25 °C) before centrifugation at 12,000 rpm for 15 min. The supernatant (500 mL) was transferred into a sterilized Eppendorf tube and mixed with isopropanol at a 1:1 volume ratio, incubated for 10 min at room temperature, and centrifuged again for 10 min. The supernatants were removed, resuspended in 75% ethanol in diethyl pyrocarbonate (DEPC)-treated water, and centrifuged for 10 min. After centrifugation, the supernatants were removed again. The precipitated pellets were air-dried and then dissolved in DEPC-treated water and stored at −80 °C for future use.

#### RT-qPCR Analysis

For the RT-qPCR analysis, RNA (20 ng) was prepared for each SeNP concentration group. Thereafter, RNA (20 µL) was reverse transcribed into cDNA using a thermal cycler (T100 Thermal Cycler; BIO-RAD, Berkeley, CA, USA). The mixture contained the sample RNA, DEPC-treated water, 10× DNase buffer, DNase I, EDTA, Oligo dT, dNTP, 5× Prime Script buffer, Prime Script, and TE buffer (10 mM Tris-HCl (pH 8.0)/0.1 mM EDTA). After preparing the cDNA sample, the RT-qPCR analysis was performed using a master mix that consisted of SYBR qPCR Mix (140 μL; THUNDERBIRD SYBR RT-qPCR Mix; TOYOBO, Osaka, Japan), forward (0.56 μL) and reverse (0.56 μL) primers of the desired gene marker, and Milli-Q Water (110.88 μL). Subsequently, the master mix (18 μL) was pipetted into LightCycler capillaries along with the transcribed cDNA sample (2 μL) and centrifuged (LC Carousel Centrifuge 2.0; Roche, Basel, Switzerland). The results were analyzed using LightCycler 2.0 (Roche, Switzerland) and its software program [25].

### 2.5. Preparation for Antimicrobial Assay

A stock of *P. gingivalis* (ATCC 33277) was received from the Department of Oral Implantology, Tokyo Medical and Dental University. The bacteria were propagated and streaked using a sterile loop onto a Vital Media Brucella HK Agar RS (Kyokuto Pharmaceutical Industrial Co., Ltd., Tokyo, Japan) and anaerobically incubated using the AnaeroPack System (Mitsubishi Gas Chemical, Tokyo, Japan) at 36 °C with 4.6% of CO_2_ for 96 h. A few colony units of *P. gingivalis* from the plate were inoculated into a test tube of anaerobic bacterial culture medium (ABCM) (Eiken Chemical Co., Ltd., Tokyo, Japan), resulting in a density of 2.0 × 10^7^ colony forming units per millimeter (CFU/mL), and anaerobically incubated for another 96 h. After the incubation period, a standard curve correlating optical density and bacterial concentration was created based on a serial dilution of the incubated bacteria. The absorbance was measured at a wavelength of 600 nm (OD_600_) using a DU-64 spectrophotometer (Beckman Coulter, Brea, CA, USA).

#### 2.5.1. Antimicrobial Activity Assay

A few colony units of *P. gingivalis* from the culture agar were inoculated into a glass test tube of ABCM (OD_600_ = 0.15) and anaerobically incubated with the AnaeroPack System. Bacterial solution (100 μL) was inoculated together with SeNPs at various concentrations (900 μL) in glass test tubes and incubated for 96 h. Blank solutions for the control group were prepared with ABCM without bacteria and without SeNPs. Blank solutions for the experimental groups were prepared by adding SeNPs of each concentration into ABCM without bacteria. After the indicated incubation period, 130 μL of solution was removed from well-vortexed bacteria/SeNP solution and placed into a cuvette for OD_600_ measurements using a Du-64 spectrophotometer (Beckman Coulter, Brea, CA, USA). The measured optical density value was extrapolated to bacterial concentration (CFU/mL) using the standard curve initially created with the DU-64 spectrophotometer.

#### 2.5.2. Minimal Bactericidal Concentrations

Samples with each SeNP concentration from the previous tests were inoculated into a Vital Media Brucella HK Agar RS to observe the bactericidal effect of the lowest concentration of nanoparticles. A sample volume of 100 µL was inoculated and anaerobically incubated with the AnaeroPack System for 72 h. Blank solutions were prepared with ABCM without bacteria and without SeNPs.

### 2.6. Statistical Analysis

Statistical analyses were conducted using SPSS 26.0 (IBM, Armonk, NY, USA) and Prism 8.0 (GraphPad Software, La Jolla, CA, USA). The one-way ANOVA (post hoc multi-comparisons with Tukey test) was used for data comparison. All data are presented as mean ± standard deviation. Results with *p* < 0.05 were considered statistically significant.

## 3. Results

### 3.1. Characteristics of SeNPs: TEM Image and Zeta Potential

The TEM image (Figure 1) of the SeNPs displays monodispersed uniform spherical shape with an average size congruent with NanoShel’s specification of 70 nm (Figure 1a,b). The Zetasizer Nano ZS also revealed the size of the SeNP to be 72.75 nm. Additionally, NanoShel specified the purity of the SeNPs to be 99.9%. The zeta potential of the SeNP in water was −21 mV. The SeNP in MEM + 10% FBS the zeta potential was shown to be −9.95 mV.

### 3.2. Proliferation and Cell Viability CCK-8 Assay

The proliferation of MC3T3 cells treated with SeNPs at 2, 4, 8, 16, 32, 64, 128, 256, 512, 1024, and 2048 ppm was determined at three different time periods. The number of cells that were viable in proliferation, determined using the colorimetric assay of CCK-8, revealed no significant levels of toxicity at all SeNP concentrations on days 3 and 5 (Figure 2). A decrease in viable proliferation as concentration increased was observed on day 7, although not statistically significant.

### 3.3. ALP Assay and Alizarin Red Staining to Analyze the Effect of SeNPs

The ALP assay was conducted to investigate the effect of SeNPs concentration on osteogenic differentiation (Figure 3a–c). Cells were treated with SeNPs at 2, 4, 8, 16, 32, 64, 128, 256, 512, 1024, and 2048 ppm in osteogenic induction medium for 3, 5, and 7 days. The relative ALP activity at SeNP concentrations of 2–32 ppm was not significantly affected compared with that of the control at all three time points. Concentrations of 256 ppm and above showed a significant decrease in relative ALP activity (*p <* 0.001), indicating that the influence of SeNPs on osteogenic differentiation is concentration-dependent.

Alizarin red staining was performed to observe the calcium deposition of MC3T3 cells treated with SeNPs at different concentrations at day 21 (Figure 3d). SeNPs at concentrations of 2, 4, and 8 ppm showed staining comparable with the control. SeNPs at concentrations of 16 and 32 ppm gradually showed a decrease in staining, indicating a low level of calcification. At concentrations of 64 ppm and higher, there was no calcification. This indicates that higher concentrations of SeNPs inhibit the mineralization and calcification processes.

### 3.4. Effect of SeNPs on the Osteoblastic Differentiation Pathway (RT-qPCR Analysis)

SeNP concentrations between 2 to 64 ppm significantly upregulated the gene expression of Osx, Runx2, Smad 1, and BMPR2, whereas gene expression of OCN, Smad 3, BMP2, and ALP was comparable to that of the control (Figure 4). This indicates that the concentrations of 2 to 64 ppm did not negatively affect the osteoblastic differentiation pathway. SeNP concentrations of 128 ppm and above were shown to significantly downregulate the expression of OCN, Osx, Smad 1, and ALP, whereas Smad 3 expression was significantly upregulated. SeNPs at high concentrations seem to affect the osteoblastic differentiation pathway.

### 3.5. Antimicrobial Effect of SeNPs

The in vitro antimicrobial assay of SeNPs was performed using *P. gingivalis*, a gram-negative rod-shaped anaerobic pathogenic bacterium. The antimicrobial effect of SeNPs against *P. gingivalis* was clearly concentration-dependent (Figure 5). Concentrations of 4 ppm and above all showed a significant reduction in CFU counts. An SeNP concentration of 4 ppm corresponded to 2.92 × 10^7^ ± 0.023 CFU/mL compared with 3.91 × 10^7^ ± 0.019 CFU/mL for the control, resulting in a significant difference of 9.90 × 10^6^ CFU/mL (*p <* 0.004). An SeNP concentration of 2048 ppm resulted in 47,900 ± 0.053 CFU/mL (*p <* 0.001), indicating a strong inhibitory effect, as the cell density of the bacteria was similar to that before incubation. Additionally, tests with *P. gingivalis* revealed that the SeNP concentrations in this study were not enough to kill a bacterium over a fixed period of time, provided that the highest concentration of SeNP in the minimal bactericidal concentration test did not show full transparency to that of blank as shown in Figure 5b,c, indicating a certain degree of bacterial presence.

## 4. Discussion

We evaluated the concentration-dependence of SeNPs on the cytotoxicity and osteoblastic differentiation of the MC3T3 cell line, as well as the antimicrobial effect against *P. gingivalis*. This work aimed to provide a foundation for future applications of SeNPs on dental implants.

### 4.1. SeNP Characteristics

TEM analysis determined the SeNPs size and morphology as 70 nm and spherical in shape. Nanoparticles within the size range of 50 and 100 nm have a higher cellular uptake compared to larger sized nanoparticles [26]. The spherical shape of nanoparticles also facilitates internalization [27]. In the addition of SeNP into a medium, sonication was used to separate and prevent nanoparticle aggregation. The use of sonication was required to effectively break down the powder agglomerates resulting from Van Der Waals inter-particle attractions [28]. Aggregation of nanoparticles could result from the presence of aging of nanoparticles from the initial synthesis; the interparticle distance decreases owing to increased electrostatic interaction [29]. Nanoparticle sedimentation occurred after a period of time. However, simple mixing allowed for an even distribution of the nanoparticles within the medium. The zeta potential of the SeNP in water was −21 mV, and the zeta potential in MEM + 10% FBS was −9.95 mV. With electrostatic stabilization, the zeta potential of the particles provides the repulsive force. Particles are considered stable if the zeta potential of the particles is higher than 30 mV or lower than -30 mV [30]. Multiple factors such as pH can affect the zeta potential. At physiological pH and electrolyte concentrations, the zeta potential of the particles is not high enough and unstable, and thus nanoparticles agglomerate [30]. The higher stability of −21mV of SeNP in distilled water could be attributed to the content of the medium. Samples in a medium with salt would be more practical since it is more similar to the condition of cell culture mediums [31]. D. Lochmann et al. identified the content of NaCl or salt mixture in mediums affected the surface charge of particles. Medium with a higher concentration of NaCl aggregated nanoparticles at a quick rate, causing the particles to be less stable. This could explain the lower stability of the SeNPs in MEM + 10% FBS due to the salt content of the MEM medium being much greater than distilled water [31,32,33].

### 4.2. Effects of SeNPs on Cell Proliferation

The proliferation of MC3T3 cells on days 3, 5, and 7 was unchanged in the presence of the SeNPs regardless of concentration. This suggests that the proliferation of MC3T3 is not affected by SeNPs. However, noticing the decreasing trend of proliferation rate from day 7, SeNPs may have a time- and concentration-dependent effect on cell proliferation. A possible explanation of such an outcome could be attributed to the direct contact of SeNP with the cells and exposure to ROS [34] A longer study period may show cytotoxicity at high SeNP concentrations. This assumption is based on the results of Alizarin Red staining of cells cultured for 21 days. The cells cultured for 21 days for ALP and Alizarin red staining were viable.

### 4.3. Effects of SeNPs on Osteoblastic Differentiation

To understand the effect of different concentrations of SeNPs on ALP, Alizarin red staining, and osteogenesis differentiation, we tested the selected gene markers (Table 1) to observe the influence of SeNPs on the TGF-β and BMP-2 Smad-dependent signaling pathways. We analyzed the gene expression on day 7 as the expression is relatively more significant owing to the relevance of genes to mineralization, whereas the analysis on day 3 provided the observation of changes in gene expression over time.

BMP-2 is a requisite factor and promoter of osteoprogenitor proliferation, early differentiation, and osteoblastic lineage commitment, which is dependent on BMPR2, a serine/threonine receptor kinase of BMP-2 signaling [44]. The signaling induction of BMPR2 causes the phosphorylation of R-Smad1 and Smad5 because of the presence of phosphate groups from BMPR2 (Figure 6). SeNPs at different concentrations evidently influence the lineage pathway. High SeNP concentrations of 128, 256, and 512 ppm significantly upregulated the expression of BMP-2. However, based on the observed calcium deposition from Alizarin red staining, high concentrations showed diminishing effects. This discrepancy could be explained by the downregulation of Smad1 and BMPR2 at SeNP concentrations of 128 ppm and above. The low expression of Smad1 resulted in the impairment of osteoblast proliferation and differentiation with a partial impairment of BMP signaling [45,46].

Runt-related transcription factor 2 (Runx2) is an essential transcription factor in the early differentiation stage regulated by BMP-2 with the interaction of Smadl and Smad5. In contrast, Smad 2 and 3 effectors are known to inhibit osteoblastic differentiation (Figure 6) [47,48,49]. The results presented in Figure 4c provided evidence of upregulation of Runx2 expression in the SeNP concentration range of 4–64 ppm (*p* < 0.001), while SeNP concentrations of 128 ppm and above did not increase the expression of Runx2. This finding could be explained by the analysis of the BMP-2 and Smad3 expression data, as Alliston et al. provided data indicating that the TGF-β–Smad pair, Smad3, possesses the ability to repress the transcriptional function of Runx2 [47]. The expression of Smad3 (Figure 4e) for every SeNP concentration was inversely proportional to that of Runx2, indicating a relationship between Smad3 and Runx2. This repression results from the direct physical association of Smad3 with Runx2 and the additional recruitment of histone deacetylases by Smad3, which represses the function of Runx2 [45,48]. Consequently, the expression of Runx2 seems to influence the expression of other osteoblastic genes [50]. Runx2 has a direct influence on osteoblast-specific genes, including ALP, osteocalcin (OCN), and Osterix [51]. The ALP gene expression pattern was in agreement with the relative ALP activity. The upregulation of ALP from 2 to 32 ppm indicates mineralization of the osteoid during bone modeling and remodeling. Even with the overexpression of BMP-2 at concentrations of 128–2048 ppm, ALP expression and mineralization were not induced, which is contrary to the finding of a previous study, which reported that ALP is indirectly controlled by BMP-2 [52].

OCN is a bone-specific protein marker that strongly indicates osteogenic maturation and mineralization [53]. Transcription factor Sp7, also known as osterix (Osx), is an indispensable factor for bone formation and mineralization. Both these markers are related to the Runx2 transcription factor and BMP-2. According to some studies, the absence of Runx2 expression leads to the absence of Osx [54,55]. However, other studies indicated that Osx expression was still induced by BMP-2 through Dlx5, a bone-inducing transcription factor that is expressed in differentiating osteoblasts [54,56]. In this study, we demonstrated the influence of SeNPs on these genetic markers. Osx expression on day 7 (Figure 4b) was significantly upregulated compared with the control (*p* < 0.002) at SeNPs concentrations of 16 to 64 ppm. Expectingly, Osx expression was similar to OCN expression. The lack of Osx expression in higher SeNPs concentrations could mainly be due to the downregulation of Smad1 [57]. Regardless of BMP-2 expression, without Smad1 expression, Osx expression was significantly hindered [46,57].

### 4.4. Antimicrobial Effects of SeNPs

In vitro antimicrobial assays with SeNPs were performed using *P. gingivalis*, which has a central role in the progressive inflammation and destruction of bone and tissue in periodontal disease, commonly in peri-implantitis. Along with two other bacteria, Treponema denticola and Tannerella forsythia, *P. gingivalis* is categorized as a highly recognized red complex bacterium involved in adult periodontal disease [58]. *P. gingivalis* alone was found in almost 90% of subgingival plaque from chronic periodontitis, with a 50–72% frequency in peri-implantitis subgingival biofilms, making this bacterium an essential etiology indicator in both periodontitis and peri-implantitis [58,59]. To demonstrate the effects of different concentrations of SeNPs on *P. gingivalis*, the bacterial suspension was inoculated along with SeNPs for 96 h, cell density was determined, and growth inhibition was observed as the concentration increased. This effect can be attributed to the reduction of sodium selenite (Na_2_SeO_3_), which is produced because of the reaction with toxic selenium dioxide [60,61,62]. The presence of the selenium compound increases the stress condition of the bacteria. Furthermore, the higher the selenium concentration, the greater the effect of Na_2_SeO_3_ on bacterial plasmid DNA [63,64]. Another contributing factor could be the area of contact, as another report on gold nanoparticles with similar effects showed antibacterial properties associated with the area of contact and proximity to *P. gingivalis* [65]. Thus, higher concentrations increase the area of contact, resulting in more antibacterial effects. These effects have been reported to observe the damaging of cell membranes of the microorganism [34,66]. SeNPs have been reported to generate a high level of ROS providing the anti-cancer properties of the nanoparticles. Additionally, few studies have reported ROS also to possess antibacterial properties [66]. This investigation provides a proof of concept for the use of SeNPs as an effective antimicrobial coating on implants. Currently, the use of non-surgical treatments for peri-implantitis cannot decrease the number of bacterial pathogens, and thus surgical protocols are required for proper decontamination and reduction in the bacterial count, followed by a maintenance phase to maintain a healthier state [67]. SeNPs may inhibit the growth of bacteria without the need for an additional surgical procedure, while potentially maintaining a decreased number of bacteria to prevent peri-implantitis. An in vivo study would be required to review this in-depth.

## 5. Conclusions

In this study on SeNPs, we assessed their osteoblastic differentiation effects and antibacterial properties, providing insights into the effect of SeNPs over a wide concentration range on osteoblastic differentiation and *P. gingivalis* activity. Our data suggest that SeNPs at low doses (4–16 ppm) are beneficial against pathogenic conditions while showing no obvious effect on osteoblastic differentiation in an in vitro scenario. With regard to the collected data and the limitations of this study, further studies should be investigated to determine the mechanism of such an effect. Furthermore, in vivo studies should be conducted to better understand the effect of the SeNPs at these concentrations on the body as a whole.

## Figures and Tables

**Figure 1 nanomaterials-12-01850-f001:**
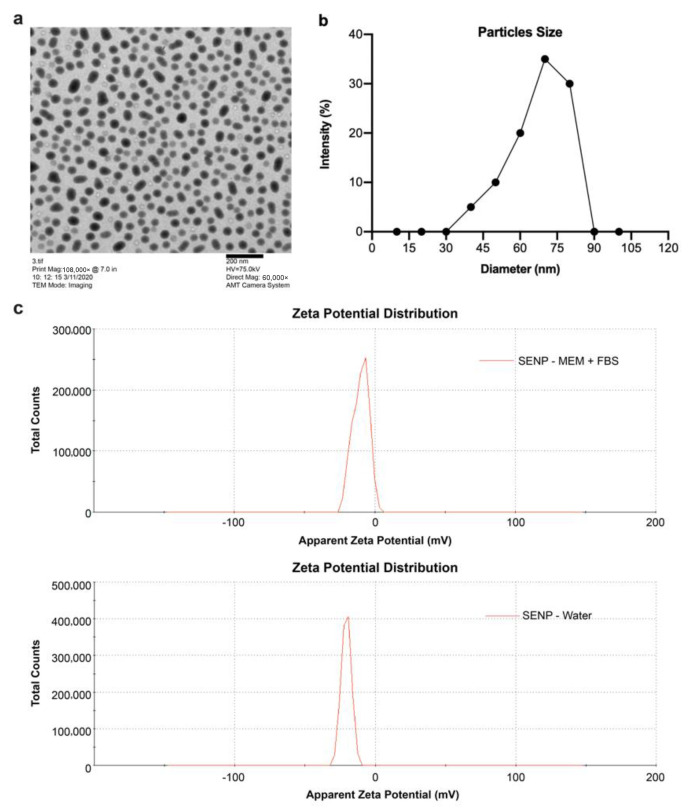
(**a**) Transmission electron microscopy (TEM) scan (scale bar: 200 nm) of selenium nanoparticles (SeNPs) at a magnification of 60,000× *g* and voltage of 75.0 kV. Nanoparticles were mostly segregated from one another and were nearly spherical. (**b**) Nanoparticle average size distribution. (**c**) Zeta potential of SeNPs at 25 °C.

**Figure 2 nanomaterials-12-01850-f002:**
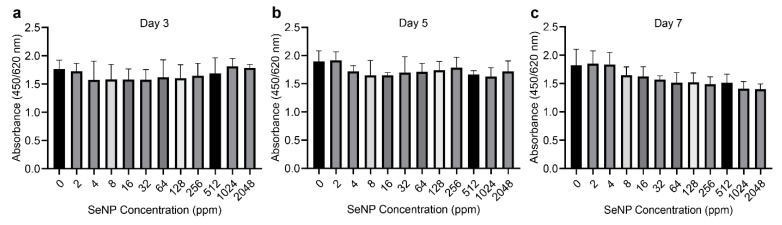
Effect of selenium nanoparticles (SeNPs) on cell viability on (**a**) day 3, (**b**) day 5, and (**c**) day 7. MC3T3 cells were cultured together with SeNPs for up to 7 days. Cell viability was measured using the Cell Counting Kit-8 (CCK-8) after 60 min of culture in CCK-8 solution. From days 3 to 7, an apparent decrease in cell viability could be observed, but it was not significantly different from that of SeNPs of 0 ppm (=mg/L) (control group). SeNPs concentration did not significantly affect cell viability. Data are expressed as mean ± standard deviation, *n* = 4. (*: *p <* 0.05, **: *p <* 0.01).

**Figure 3 nanomaterials-12-01850-f003:**
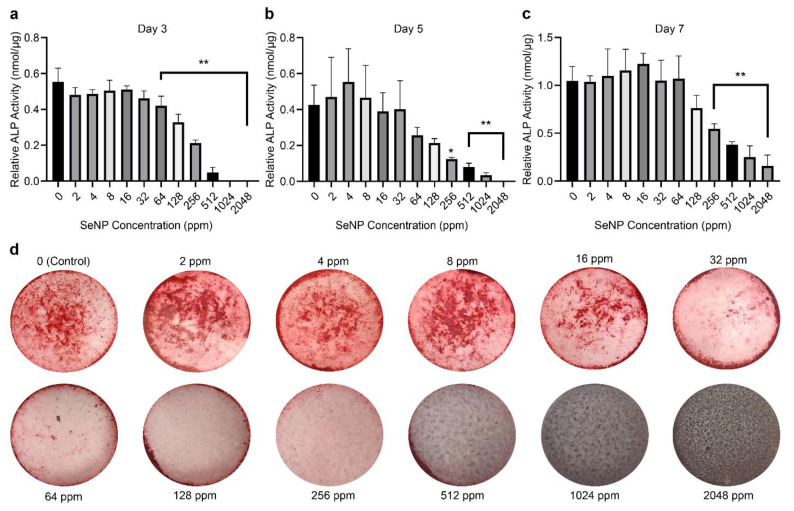
Effect of selenium nanoparticles (SeNPs) on relative alkaline phosphatase (ALP) activity on (**a**) day 3 (**b**) day 5, and (**c**) day 7. Relative ALP activity was calculated based on ALP and bicinchoninic acid assays. (**d**) Alizarin red staining after 21 days. Samples with SeNP concentrations of 2–16 ppm showed calcification comparable with the control, whereas those with SeNP concentrations of 32 ppm and above showed zero to low levels of calcification. Data are expressed as mean ± standard deviation, *n* = 4. (*: *p <* 0.05, **: *p <* 0.01).

**Figure 4 nanomaterials-12-01850-f004:**
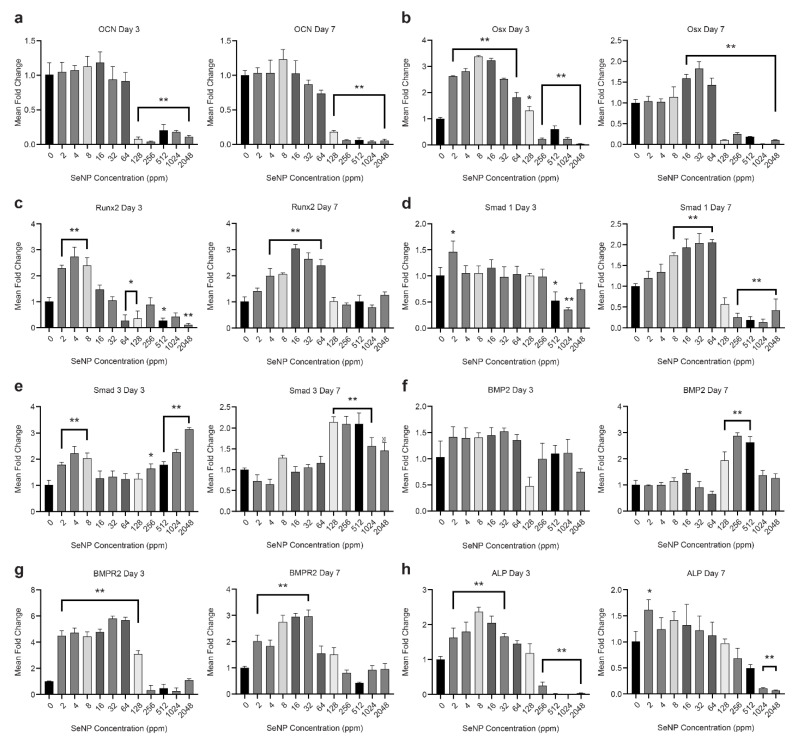
Gene expression analysis using reverse transcriptase quantitative polymerase chain reaction on days 3 and 7. The expression levels of the tested gene markers were normalized based on hypoxanthine-guanine phosphoribosyl transferase (HPRT) expression. Primer used in the analysis are as follows: (**a**) osteocalcin, (**b**) osterix, (**c**) runt-related transcription factor 2, (**d**) Smad 1, (**e**) Smad 3, (**f**) bone morphogenetic proteins 2, (**g**) bone morphogenetic proteins receptor 2, and (**h**) alkaline phosphatase. Data are expressed as mean ± standard deviation, *n* = 3. (*: *p <* 0.05, **: *p <* 0.01).

**Figure 5 nanomaterials-12-01850-f005:**
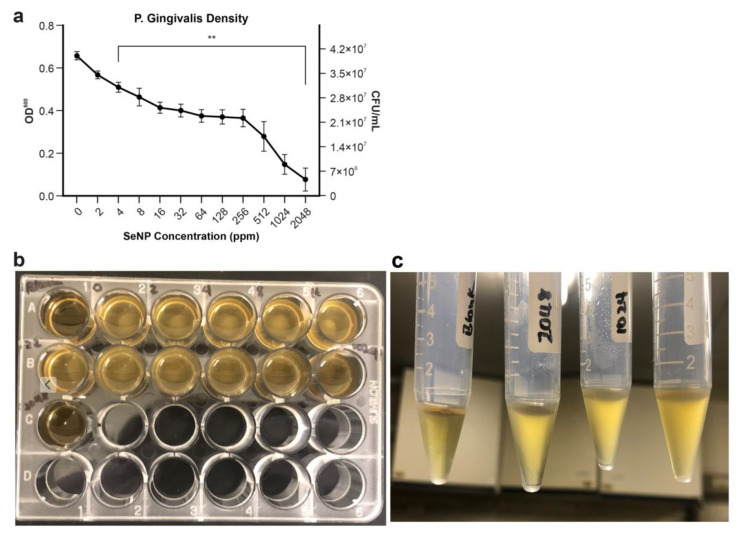
Effect of selenium nanoparticle (SeNP) concentration on *P. gingivalis* growth. SeNPs were added to *P. gingivalis* solution, and the resultant solution was incubated for 96 h before examination. (**a**) Standard curve of optical density at 600 nm (OD_600_) was used to extrapolate the colony forming units per millimeter (CFU/mL) data. Data are expressed as mean ± standard deviation, n = 3. (*: *p* < 0.05, **: *p* < 0.01). (**b**) Minimal bactericidal concentration of SeNP on *P. gingivalis* (OD: 0.15). Blank without *P. gingivalis* (Well A1). *P. gingivalis* without SeNP (Well A2). SeNP concentration of 2048 (Well C1) showing slight evidence of bacteria with close similarity in transparency to blank. (**c**) Transparency of blank in comparison to SeNP concentrations.

**Figure 6 nanomaterials-12-01850-f006:**
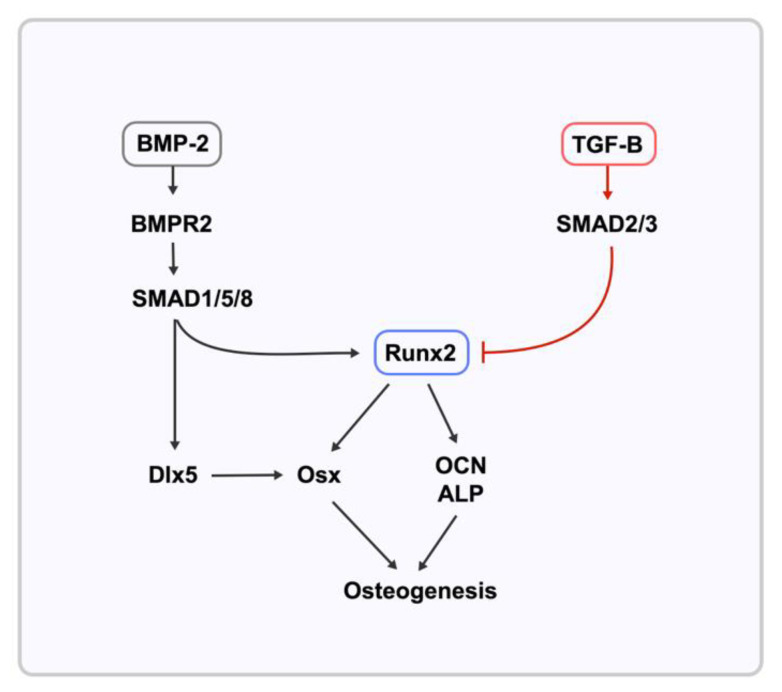
Pathway of gene expression tested against SeNPs.

**Table 1 nanomaterials-12-01850-t001:** Primers used in the gene expression analysis with reverse transcriptase quantitative polymerase chain reaction.

Gene Marker	Primer	Sequence (5′–3′)	Accession Number	References
HPRT ^†^	forward reverse	TATGTCCCCCGTTGACTGAT CTTTGCTGACCTGCTGGATT	NM_013556.2	[24,35]
OCN ^‡^	forward reverse	GCAATAAGGTAGTGAACAGACTCC GTTTGTAGGCGGTCTTCAAGC	NM_007541	[36]
Osx ^§^	forward reverse	CCCACCTAACAGGAGGATTT CACTGGAATGGAGTGAAACC	NM_130458.3	[37]
Runx2 ^¶^	forward reverse	TGCTATTGCCCAAGATTTGC GAGGGGGAAATGCCAAATAA	N/A	[37,38]
ALP ^††^	forward reverse	GGGCGTCTCCACAGTAACCG ACTCCCACTGTGCCCTCGTT	N/A	[39,40]
Smad1	forward reverse	CTGAAGCCTCTGGAATGCTGTG CAGAAGGCTGTGCTGAGGATTG	NM_008539	[41]
Smad3	forward reverse	GCTTTGAGGCTGTCTACCAGCT GTGAGGACCTTGACAAGCCACT	NM_016769	[42]
BMP-2 ^‡‡^	forward reverse	GAAATCTCCAAGTGCCCAAA GGTGTTGAGAAGCCTGAAGC	N/A	[39,40]
BMPR2 ^§§^	forward reverse	AGAGACCCAAGTTCCCAGAAGC TCTCCTCAGCACACTGTGCAGT	NM_007561	[43]

^†^ Hypoxanthine-guanine phosphoribosyl transferase; ^‡^ osteocalcin; ^§^ osterix; ^¶^ Runt-related transcription factor 2; ^††^ alkaline phosphatase; ^‡‡^ bone morphogenetic protein 2; ^§§^ bone morphogenetic protein receptor 2. Accession numbers are from RefSeq (https://www.ncbi.nlm.nih.gov/refseq/, accessed on 23 November 2021).

## Data Availability

The data that support the findings of this study are available from the corresponding authors upon reasonable request.

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
