# Peer review of "An In Vitro Evaluation of Selenium Nanoparticles on Osteoblastic Differentiation and Antimicrobial Properties against Porphyromonas gingivalis"

_nanomaterials, 2022, doi:10.3390/nano12111850_

Round 1

Reviewer 1 Report

The current manuscript aims to perform in vitro evaluation of selenium nanoparticles on osteoblastic differentiation and antimicrobial properties against porphyromonas gingivalis for surface modification of the dental implants. Although the topic is significant in the field of development of metallic nanoparticles for antibacterial dental biomaterials, there are several issues that definitely require the authors’ attention to improve the quality of this particular manuscript before further consideration for publication in a high-quality journal “Nanomaterials”.

Specific comments:

  1. The manuscript title was presented as “An In-Vitro Evaluation of Selenium Nanoparticles on Osteoblastic Differentiation and Antimicrobial Properties Against Porphyromonas gingivalis for Surface Modification of the Dental Implants”. However, it seems unsuitable due to the absence of any scientific evidences relevant to surface modification of the dental implants. In my opinion, this work simply focused on the in vitro study of selenium nanoparticles. In order to better emphasize the research core, the authors are highly recommended to modify the manuscript title.
  2. As stated by the authors, “In recent decades, nanoparticles have been extensively studied owing to their therapeutic properties”. However, this important claim is not supported by any documented references. In fact, some recent reviews involving the investigation of various therapeutic nanoparticles for disease treatment have been reported (Please refer to the following papers: Chemical Engineering Journal 2022;435:134970 & Molecules 2022;27:392). In order to balance the reference viewpoint and attract more attention from audiences, the authors are highly recommended to consider the inclusion of this relevant publication in the reference list to enrich the article content.
  3. As shown in Figure 1c, the zeta potential of the SeNP in water was -21 mV, and the zeta potential in MEM + 10% FBS was -9.95 mV. One possible explanation for this observation is that the surface charges carried by the nanoparticles are screened by medium components [please refer to the following paper: Journal of Controlled Release 2004;100:411-423]. The authors are highly recommended to consider the aforementioned article and give reasonable explanation about the change of surface charge density of SeNP samples in different solutions (i.e., deionized water versus physiological buffer) in the Discussion section.
  4. As stated by the authors, a decrease in viable proliferation as concentration increased was observed on day 7, although not statistically significant (Figure 2). However, the audiences are curious about why the increase in SeNP concentration may cause the decrease in metabolic activity of MC3T3 cells? Please specify the underlying mechanism.
  5. As stated by the authors, tests with P. gingivalis revealed that the SeNP concentrations in this study were not bactericidal, as colony formation was observed in the agar plates (Figure 5). The authors should provide relevant evidences to support this important finding.
  6. More importantly, the authors are highly recommended to clarify the antimicrobial mechanism of the SeNP by performing a comprehensive investigation.
  7. In my opinion, the present study demonstrated the effect of SeNP concentration on cell proliferation (rather than cytotoxicity). In order to confirm the safety of SeNP, the authors are highly recommended to perform live/dead bioassays.

Author Response

Reviewers’ Comments and Authors Response

Paper ID: nanomaterials-1729220 

Paper title: An In-Vitro Evaluation of Selenium Nanoparticles on Osteoblastic Differentiation and Antimicrobial Properties Against Porphyromonas gingivalis for Surface Modification of the Dental Implants.

Authors: Jason Hou, Yukihiko Tamura, Hsin-Ying Lu, Yuta Takahashi, Shohei Kasugai, Hidemi Nakata, Shinji Kuroda

The authors would like to thank the area editor and the reviewers for their precious time and invaluable comments. We have carefully addressed all the comments. The corresponding changes and refinements made in the revised paper are summarized in our response below in blue.

Reviewer 1:

Comments and Suggestions for Authors

The current manuscript aims to perform in vitro evaluation of selenium nanoparticles on osteoblastic differentiation and antimicrobial properties against porphyromonas gingivalis for surface modification of the dental implants. Although the topic is significant in the field of development of metallic nanoparticles for antibacterial dental biomaterials, there are several issues that definitely require the authors’ attention to improve the quality of this particular manuscript before further consideration for publication in a high-quality journal “Nanomaterials”.

Specific comments:

  1. The manuscript title was presented as “An In-Vitro Evaluation of Selenium Nanoparticles on Osteoblastic Differentiation and Antimicrobial Properties Against Porphyromonas gingivalis for Surface Modification of the Dental Implants”. However, it seems unsuitable due to the absence of any scientific evidences relevant to surface modification of the dental implants. In my opinion, this work simply focused on the in vitro study of selenium nanoparticles. In order to better emphasize the research core, the authors are highly recommended to modify the manuscript title.

It was mentioned that the title seems unsuitable due to the absence of any scientific evidences relevant to surface modification of the dental implants. The goal of the paper was to observe the nanoparticle effect and to be utilized for surface modification for dental implants. From the reviewer’s raised points, we understand the potential misinterpretation about the title and the content of the article. Thus, we have changed the title of the paper to “An In-Vitro Evaluation of Selenium Nanoparticles on Osteoblastic Differentiation and Antimicrobial Properties Against Porphyromonas gingivalis” to better emphasize the core of our research on what was investigated.

Essentially, we have only removed “for Surface Modification of the Dental Implants”. After careful consideration of the title, we find that the new title correctly denotes the entirety of the research.

2. As stated by the authors, “In recent decades, nanoparticles have been extensively studied owing to their therapeutic properties”. However, this important claim is not supported by any documented references. In fact, some recent reviews involving the investigation of various therapeutic nanoparticles for disease treatment have been reported (Please refer to the following papers: Chemical Engineering Journal 2022;435:134970 & Molecules 2022;27:392). In order to balance the reference viewpoint and attract more attention from audiences, the authors are highly recommended to consider the inclusion of this relevant publication in the reference list to enrich the article content.

We thank the reviewer for the inclusion of the journal articles that have been provided. We have edited the statement reflected on the paper, to be more specific on the studies of nanoparticles and progression of its application to medicine for therapeutic properties. Moreover, as suggested, after reading the articles we included the 2 articles as a reference to support our statement. Additionally, though it was not asked by the reviewer, we felt it essential to add 2 additional references to support the statement that the study on nanoparticles has been widely or extensively done. We hope this clears up any unsupported claim which may cause some misunderstandings.

3. As shown in Figure 1c, the zeta potential of the SeNP in water was -21 mV, and the zeta potential in MEM + 10% FBS was -9.95 mV. One possible explanation for this observation is that the surface charges carried by the nanoparticles are screened by medium components [please refer to the following paper: Journal of Controlled Release 2004;100:411-423]. The authors are highly recommended to consider the aforementioned article and give reasonable explanation about the change of surface charge density of SeNP samples in different solutions (i.e., deionized water versus physiological buffer) in the Discussion section.

We thank the reviewer for providing an invaluable reference for the paper on the zeta potential. This article has led to other articles related to this topic. It has caught our attention that the studies experimented on different kinds of medium which resulted in different surface charges due to the content of the medium. The article provided referenced to 2 other articles that gave an explanation on the potential cause of different surface charge is due to the salt content of the medium. The concentration of the salt content results in the instability of the nanoparticles and increases the rate of aggregation.

We acknowledge that the discussion in our paper related to the work on zeta potential was incomplete. We have included the above explanation in the paper and added several bibliographical references, as follows:

  1. Lochmann, Dirk & Vogel, Vitali & Weyermann, Jörg & Dinauer, N & v. Briesen, Hagen & Kreuter, Jörg & Schbert, D & Zimmer, Andreas. Physicochemical characterization of protamine-phosphorothioate nanoparticles. Journal of microencapsulation. 21. 625-41. doi:10.1080/02652040400000504.
  2. Lochmann D, Weyermann J, Georgens C, Prassl R, Zimmer A. Albumin-protamine-oligonucleotide nanoparticles as a new antisense delivery system. Part 1: physicochemical characterization. Eur J Pharm Biopharm. 2005;59(3):419-429. doi:10.1016/j.ejpb.2004.04.001

  3. Weyermann J, Lochmann D, Zimmer A. Comparison of antisense oligonucleotide drug delivery systems. J Control Release. 2004;100(3):411-423. doi:10.1016/j.jconrel.2004.08.027

4. As stated by the authors, a decrease in viable proliferation as concentration increased was observed on day 7, although not statistically significant (Figure 2). However, the audiences are curious about why the increase in SeNP concentration may cause the decrease in metabolic activity of MC3T3 cells? Please specify the underlying mechanism.

The reviewer and audience have brought forth a question which is important and so we have included the possible explanation in the paper in the discussion.

There are many factors that can affect the decrease of metabolic activity. However, in relation to the increase in concentration, there is a strong possibility the cause may be due to the direct interaction between the nanoparticles and cells attributed to the increase of direct contact. On a cellular level, nanoparticles interact with the cell membranes leading to the absorption or compromising of integrity. This in turn leads to the interaction of the nucleus, lysosomes and mitochondria. Concurrently, high levels of reactive oxygen species (ROS) can cause such damage.

However, these mechanisms have not been found to be mutually exclusive, making it complex to determine the exact mechanism for nanoparticle cytotoxicity.

5. As stated by the authors, tests with P. gingivalis revealed that the SeNP concentrations in this study were not bactericidal, as colony formation was observed in the agar plates (Figure 5). The authors should provide relevant evidences to support this important finding.

We thank the reviewer for bringing our attention to the matter by providing relevant evidence to the finding.

There are two points we would like to address.

  1. From the minimal bactericidal concentration assay, we concluded “SeNP concentrations in this study were not bactericidal”. This statement is a misrepresentation of the minimal bactericidal concentration assay. Hence, the more correct representation of this assay would be the following sentence “Additionally, tests with P. gingivalis revealed that the SeNP concentrations in this study were not enough to kill the bacterium over a fixed period”.

This is an important statement which we have corrected in our paper thanks to the observation of the reviewer.

  1. Regarding relevant evidence, because it is true that we did not provide images of colony formed on agar place, the statement “as colony formation was observed in the agar” was modified to a statement with the same implication but redirected to the evidence already provided in Figure 5.

Both Figure 5a and 5b showed the highest concentration of the SeNP (2048ppm) presented with a degree of bacteria (significantly less degree in comparison to the degree of 2ppm SeNPs). From our knowledge, the minimal bactericidal concentration assay is significant in providing evidence on whether or not each SeNPs concentration possess the ability to kill all bacterium measured.

From Figure 5a, the antimicrobial investigations identified that the bacterial counts were decreased significantly but were not completely absent in the presence of high concentration of the nanoparticles.

In addition to this, the minimal bactericidal concentration assay in Figure 5b, supports the idea that the highest concentration of 2048 ppm is not the lowest concentration to kill all bacterium as the medium’s transparency level does not reach the level of blank. This signifies to us that the current level of nanoparticles concentration has some inhibitory effects to the bacteria’s growth evident by the decrease in bacterial count, but not enough to eliminate all the bacteria. To help better visualize the difference in transparency, we have added an image from the same investigation into the paper as Figure 5c. We hope this image can provide the reviewer and audiences with a clearer contrast between the 2048 ppm SeNPs and blank.

6. More importantly, the authors are highly recommended to clarify the antimicrobial mechanism of the SeNP by performing a comprehensive investigation.

While we agree that determining the mechanism is valuable, we would like to provide some possible explanation to the mechanism. Similar to nanoparticles against cells, the increase of concentration allows for an increase of direct contact to the cell membrane of the microorganism. In one study in Environmental Research (doi:10.1016/j.envres.2020.110630), it was observed through SEM that after contact of SeNPs to bacteria, damages and pits were observe on the cell membranes which likely lead to the death of the cell. SeNPs have been reported, by other articles, to induce high level of ROS which attributes to the anti-cancer properties. Few papers (stated by the article doi:10.1016/j.envres.2020.110630), have reported that ROS can also function in the antimicrobial process. In relation, we explained in our paper that the SeNPs were synthesis by reducing sodium selenite (Na2SeO3(H2O)5) with ascorbic acid. Sodium selenite is a toxic compound and with increased concentration, it is safe to say there will be an increase of the compound, leading to the stress condition of the bacteria.

As we mentioned earlier, the mechanism is a valuable addition to the paper, however we would like to kindly point out that this paper is focused on the effect of the nanoparticle at different concentration levels for observation and we evaluated the performance of the nanoparticles against the specific cell lines and bacteria; the underlying mechanism for the effect of the nanoparticle is a complementary work, but not at all our focus in this study. Nevertheless, we believe the observation of the reviewer is important, and we plan to study the antimicrobial mechanism of the nanoparticles in a more detailed manner in the future.

7. In my opinion, the present study demonstrated the effect of SeNP concentration on cell proliferation (rather than cytotoxicity). In order to confirm the safety of SeNP, the authors are highly recommended to perform live/dead bioassays.

We understand the reviewer's opinion regarding the cytotoxicity. The cytotoxicity is an important experiment and so we want to make sure effort was made to verify the significance of CCK-8 as a cytotoxicity assay. From the certificate of analysis, stated by Sigma-Aldrich, CCK-8 (or WST assay) is for the quantification for the cell viability in proliferation and cytotoxicity assay. Moreover, in order to further confirm this claim, recent papers with cytotoxicity investigations were reviewed to support the findings.

Reference:

  1. Nature 576, 452–458 (2019). https://doi.org/10.1038/s41586-019-1665-6
  2. Nanomaterials 2022, 12, 569. https://doi.org / 10.3390 /nano12030569
  3. Scientific Reports | 7: 3827 | DOI:10.1038/s41598-017-04229-z
  4. Nanomaterials 2021, 11, 617. https: / /doi.org/10.3390 /nano11030617
  5. Commun., 1999,36, 47-50. DOI https://doi.org/10.1039/A809656B
  6. Journal of Controlled Release 2004;100:411-423

The papers mentioned above were used as consideration and support for the significance of CCK-8/WST Assay. The cytotoxicity assay conducted for their respective studies coincides with the use of proliferation assay in cell viability. This includes MTT assays and CCK-8/WST assays. Addition to these two tests, it was also mention in reference (Nanomaterials 2022, 12, 569) that an ALP and Alizarin red stain were indirect cytotoxicity analysis for their nanomaterials. Our paper also investigated ALP and Alizarin Red staining primarily to observe the calcification of osteoblastic differentiation. However, we believe the survivability of the cells incubated with SeNPs were viable after prolonged incubation periods, consequently allowing for the completion of the ALP and Alizarin Red assay. Nevertheless, we understand and acknowledge the opinion and view of the reviewer and plan to investigate the cell viability in a more detailed manner in the future.

Reviewer 2 Report

This paper reports an in vitro evaluation of the antimicrobial properties of Selenium Nanoparticles for their potential use in Dental Implants. The paper includes interesting results with suitable experimental design, data analysis and discussion.

Introduction

  • It is important to remark the main novelty of this work.

Conclusions

  • Further possible studies related to the present work should be mentioned.

References

  • Include more references from the journal.

Author Response

Reviewers’ Comments and Authors Response

Paper ID: nanomaterials-1729220 

Paper title: An In-Vitro Evaluation of Selenium Nanoparticles on Osteoblastic Differentiation and Antimicrobial Properties Against Porphyromonas gingivalis for Surface Modification of the Dental Implants.

Authors: Jason Hou, Yukihiko Tamura, Hsin-Ying Lu, Yuta Takahashi, Shohei Kasugai, Hidemi Nakata, Shinji Kuroda

The authors would like to thank the area editor and the reviewers for their precious time and invaluable comments. We have carefully addressed all the comments. The corresponding changes and refinements made in the revised paper are summarized in our response below in blue.

Reviewer 2

Comments and Suggestions for Authors

This paper reports an in vitro evaluation of the antimicrobial properties of Selenium Nanoparticles for their potential use in Dental Implants. The paper includes interesting results with suitable experimental design, data analysis and discussion.

Introduction

  • It is important to remark the main novelty of this work.

We thank the reviewer for kindly pointing out this point. We have explained the novelty of our work in the introduction. However, it seems we were not able to express this point across to the reviewer and the audience in an obvious manner. To prevent future misunderstandings, we have followed the reviewer’s insight and made changes to the remark.

We have made changes to the following original sentence in yellow, and newly added sentence in the color red:

In the field of orthopedics and dentistry, an ideal implant surface modification should exhibit both antibacterial and osteoblastic properties. Thus, with none or very few existing studies, we aimed to provide new data on the effects of SeNPs concentrations with experimental assessments investigated based on these ideal properties.

We believe that providing this sentence will deliver an easy identification on the purpose of our paper, to provide the effects of SeNPs concentration on important properties forementioned above.

Conclusions

  • Further possible studies related to the present work should be mentioned.

We thank the reviewer for mentioning this. Further possible studies such as investigating the possible mechanisms were added to the conclusion. We believe investigating the mechanism and an in vivo study for the future to be the next subsequent step as it would benefit us and future researchers interested in our study with a direction closer to possible medical application.

References

  • Include more references from the journal.

We have included additional references for the introduction as well as the discussion of the paper. With the insight from the reviewer, we also found it necessary to provide additional support to our points.

We have added several bibliographical references, as follows:

  1. Lochmann, Dirk & Vogel, Vitali & Weyermann, Jörg & Dinauer, N & v. Briesen, Hagen & Kreuter, Jörg & Schbert, D & Zimmer, Andreas. Physicochemical characterization of protamine-phosphorothioate nanoparticles. Journal of microencapsulation. 21. 625-41. doi:10.1080/02652040400000504.

  2. Lochmann D, Weyermann J, Georgens C, Prassl R, Zimmer A. Albumin-protamine-oligonucleotide nanoparticles as a new antisense delivery system. Part 1: physicochemical characterization. Eur J Pharm Biopharm. 2005;59(3):419-429. doi:10.1016/j.ejpb.2004.04.001

  3. Weyermann J, Lochmann D, Zimmer A. Comparison of antisense oligonucleotide drug delivery systems. J Control Release. 2004;100(3):411-423. doi:10.1016/j.jconrel.2004.08.027

  4. Nguyen, Duc & Lai, Jui-Yang. Synthesis, bioactive properties, and biomedical applications of intrinsically therapeutic nanoparticles for disease treatment. Chemical Engineering Journal. 2022; 435. 134970. doi:10.1016/j.cej.2022.134970.

  5. Zhang L, Gu FX, Chan JM, Wang AZ, Langer RS, Farokhzad OC. Nanoparticles in medicine: therapeutic applications and developments. Clin Pharmacol Ther. 2008;83(5):761-769. doi:10.1038/sj.clpt.6100400

  6. Yang T, Lee S-Y, Park K-C, Park S-H, Chung J, Lee S. The Effects of Selenium on Bone Health: From Element to Therapeutics. Molecules. 2022; 27(2):392. doi:3390/molecules27020392

  7. Zhang, Chenyang & Yan, Liang & Wang, Xin & Zhu, Shuang & Chen, Chunying & Zhao, Yuliang. Progress, challenges, and future of nanomedicine. Nano Today. 2020; 35. 101008. doi:10.1016/j.nantod.2020.101008.

  8. Henglin Zhang, Zheng Li, Chunxiao Dai, Ping Wang, Shuling Fan, Bin Yu, Yuanyuan Qu, Antibacterial properties and mechanism of selenium nanoparticles synthesized by Providencia sp. DCX, Environmental Research, 2021; 194, 110630,doi:10.1016/j.envres.2020.110630.

  9. Saifi MA, Khan W, Godugu C. Cytotoxicity of Nanomaterials: Using Nanotoxicology to Address the Safety Concerns of Nanoparticles. Pharm Nanotechnol. 2018;6(1):3-16. doi:10.2174/2211738505666171023152928

Round 2

Reviewer 1 Report

The revised version has adequately addressed most of the critiques raised by this reviewer and is now suitable for publication in "Nanomaterials".